# Reconciling "priors" & "priors" without prejudice?

**Rémi Gribonval** [*]
Inria
Centre Inria Rennes - Bretagne Atlantique
remi.gribonval@inria.fr

**Pierre Machart**
Inria
Centre Inria Rennes - Bretagne Atlantique
pierre.machart@inria.fr

## Abstract

There are two major routes to address linear inverse problems. Whereas regularization-based approaches build estimators as solutions of penalized regression optimization problems, Bayesian estimators rely on the posterior distribution of the unknown, given some assumed family of priors. While these may seem radically different approaches, recent results have shown that, in the context of additive white Gaussian denoising, the Bayesian conditional mean estimator is always the solution of a penalized regression problem. The contribution of this paper is twofold. First, we extend the additive white Gaussian denoising results to general linear inverse problems with colored Gaussian noise. Second, we characterize conditions under which the penalty function associated to the conditional mean estimator can satisfy certain popular properties such as convexity, separability, and smoothness. This sheds light on some tradeoff between computational efficiency and estimation accuracy in sparse regularization, and draws some connections between Bayesian estimation and proximal optimization.

## 1 Introduction

Let us consider a fairly general linear inverse problem, where one wants to estimate a parameter vector $z \in \mathbb{R}^D$, from a noisy observation $y \in \mathbb{R}^n$, such that $y = \boldsymbol{A}z + b$, where $\boldsymbol{A} \in \mathbb{R}^{n \times D}$ is sometimes referred to as the observation or design matrix, and $b \in \mathbb{R}^n$ represents an additive Gaussian noise with a distribution $P_B \sim \mathcal{N}(0, \Sigma)$. When $n < D$, it turns out to be an ill-posed problem. However, leveraging some prior knowledge or information, a profusion of schemes have been developed in order to provide an appropriate estimation of $z$. In this abundance, we will focus on two seemingly very different approaches.

### 1.1 Two families of approaches for linear inverse problems

On the one hand, Bayesian approaches are based on the assumption that $z$ and $b$ are drawn from probability distributions $P_Z$ and $P_B$ respectively. From that point, a straightforward way to estimate $z$ is to build, for instance, the *Minimum Mean Squared Estimator* (MMSE), sometimes referred to as *Bayesian Least Squares*, *conditional expectation* or *conditional mean* estimator, and defined as:

$$\psi_{\text{MMSE}}(y) := \mathbb{E}(Z|Y = y). \tag{1}$$

This estimator has the nice property of being optimal (in a least squares sense) but suffers from its explicit reliance on the prior distribution, which is usually unknown in practice. Moreover, its computation involves a tedious integral computation that generally cannot be done explicitly.

On the other hand, regularization-based approaches have been at the centre of a tremendous amount of work from a wide community of researchers in machine learning, signal processing, and more

---

[*]The authors are with the PANAMA project-team at IRISA, Rennes, France.

generally in applied mathematics. These approaches focus on building estimators (also called *decoders*) with no explicit reference to the prior distribution. Instead, these estimators are built as an optimal trade-off between a *data fidelity* term and a term promoting some regularity on the solution. Among these, we will focus on a widely studied family of estimators $\psi$ that write in this form:

$$\psi(y) := \underset{z \in \mathbb{R}^D}{\operatorname{argmin}} \frac{1}{2}\|y - \boldsymbol{A}z\|^2 + \phi(z). \tag{2}$$

For instance, the specific choice $\phi(z) = \lambda\|z\|_2^2$ gives rise to a method often referred to as the *ridge regression* [1] while $\phi(z) = \lambda\|z\|_1$ gives rise to the famous *Lasso* [2].

The $\ell^1$ decoder associated to $\phi(z) = \lambda\|z\|_1$ has attracted a particular attention, for the associated optimization problem is convex, and generalizations to other "mixed" norms are being intensively studied [3, 4]. Several facts explain the popularity of such approaches: a) these penalties have well-understood geometric interpretations; b) they are known to be sparsity promoting (the minimizer has many zeroes); c) this can be exploited in active set methods for computational efficiency [5]; d) convexity offers a comfortable framework to ensure both a unique minimum and a rich toolbox of efficient and provably convergent optimization algorithms [6].

## 1.2 Do they really provide different estimators?

Regularization and Bayesian estimation seemingly yield radically different viewpoints on inverse problems. In fact, they are underpinned by distinct ways of defining signal models or "priors". The "regularization prior" is embodied by the penalty function $\phi(z)$ which promotes certain solutions, somehow carving an implicit signal model. In the Bayesian framework, the "Bayesian prior" is embodied by where the mass of the signal distribution $P_Z$ lies.

**The MAP *quid pro quo*** A *quid pro quo* between these distinct notions of priors has crystallized around the notion of *maximum a posteriori* (MAP) estimation, leading to a long lasting incomprehension between two worlds. In fact, a simple application of Bayes rule shows that under a Gaussian noise model $b \sim \mathcal{N}(0, \mathbf{I})$ and *Bayesian prior* $P_Z(z \in E) = \int_E p_Z(z)dz$, $E \subset \mathbb{R}^N$, MAP estimation[1] yields the optimization problem (2) with *regularization prior* $\phi_Z(z) := -\log p_Z(z)$. By a trivial identification, the optimization problem (2) with regularization prior $\phi(z)$ is now routinely called "MAP with prior $\exp(-\phi(z))$". With the $\ell^1$ penalty, it is often called "MAP with a Laplacian prior". As an unfortunate consequence of an erroneous "reverse reading" of this fact, this identification has given rise to the erroneous but common myth that the optimization approach is particularly well adapted when the unknown is distributed as $\exp(-\phi(z))$. As a striking counter-example to this myth, it has recently been proved [7] that when $z$ is drawn i.i.d. Laplacian and $\boldsymbol{A} \in \mathbb{R}^{n \times D}$ is drawn from the Gaussian ensemble, the $\ell^1$ decoder – and indeed *any* sparse decoder – will be outperformed by the least squares decoder $\psi_{LS}(y) := \operatorname{pinv}(\boldsymbol{A})y$, unless $n \gtrsim 0.15D$.

In fact, [8] warns us that the MAP estimate is only one of the plural possible Bayesian interpretations of (2), even though it is the most straightforward one. Furthermore, to point out that erroneous conception, a deeper connection is dug, showing that in the more restricted context of (white) Gaussian denoising, for any prior, there exists a regularizer $\phi$ such that the MMSE estimator can be expressed as the solution of problem (2). This result essentially exhibits a regularization-oriented formulation for which two radically different interpretations can be made. It highlights the important following fact: the specific choice of a regularizer $\phi$ does not, alone, induce an implicit prior on the supposed distribution of the unknown; besides a prior $P_Z$, a Bayesian estimator also involves the choice of a loss function. For certain regularizers $\phi$, there can in fact exist (at least two) different priors $P_Z$ for which the optimization problem (2) yields the optimal Bayesian estimator, associated to (at least) two different losses (e.g.., the 0/1 loss for the MAP, and the quadratic loss for the MMSE).

## 1.3 Main contributions

A first major contribution of that paper is to extend the aforementioned result [8] to a more general linear inverse problem setting. Our first main results can be introduced as follows:

*Theorem* (Flavour of the main result). For *any* non-degenerate[2] prior $P_Z$, *any* non-degenerate covariance matrix $\boldsymbol{\Sigma}$ and *any* design matrix $\boldsymbol{A}$ with full rank, there exists a regularizer $\phi_{\boldsymbol{A},\boldsymbol{\Sigma},P_Z}$ such that the MMSE estimator of $z \sim P_Z$ given the observation $y = \boldsymbol{A}z + b$ with $b \sim \mathcal{N}(0, \boldsymbol{\Sigma})$,

$$\psi_{\boldsymbol{A},\boldsymbol{\Sigma},P_Z}(y) := \mathbb{E}(Z|Y = y), \tag{3}$$

is a minimizer of $z \mapsto \frac{1}{2}\|y - \boldsymbol{A}z\|_{\boldsymbol{\Sigma}}^2 + \phi_{\boldsymbol{A},\boldsymbol{\Sigma},P_Z}(z)$.

Roughly, it states that for the considered inverse problem, for any prior on $z$, the MMSE estimate with Gaussian noise is also the solution of a regularization-based problem (the converse is not true).

In addition to this result we further characterize properties of the penalty function $\phi_{\boldsymbol{A},\boldsymbol{\Sigma},P_Z}(z)$ in the case where $\boldsymbol{A}$ is invertible, by showing that: a) it is convex if and only if the probability density function of the observation $y$, $p_Y(y)$ (often called the *evidence*), is log-concave; b) when $\boldsymbol{A} = \mathbf{I}$, it is a separable sum $\phi(z) = \sum_{i=1}^{n} \phi_i(z_i)$ where $z = (z_1, \ldots, z_n)$ if, and only if, the evidence is multiplicatively separable: $p_Y(y) = \Pi_{i=1}^{n}p_{Y_i}(y_i)$.

## 1.4 Outline of the paper

In Section 2, we develop the main result of our paper, that we just introduced. To do so, we review an existing result from the literature and explicit the different steps that make it possible to generalize it to the linear inverse problem setting. In Section 3, we provide further results that shed some light on the connections between MMSE and regularization-oriented estimators. Namely, we introduce some commonly desired properties on the regularizing function such as separability and convexity and show how they relate to the priors in the Bayesian framework. Finally, in Section 4, we conclude and offer some perspectives of extension of the present work.

## 2 Main steps to the main result

We begin by a highlight of some intermediate results that build into steps towards our main theorem.

### 2.1 An existing result for white Gaussian denoising

As a starting point, let us recall the existing results in [8] (Lemma II.1 and Theorem II.2) dealing with the additive white Gaussian denoising problem, $\boldsymbol{A} = \mathbf{I}, \boldsymbol{\Sigma} = \mathbf{I}$.

**Theorem 1** (Reformulation of the main results of [8]). *For any non-degenerate prior $P_Z$, we have:*

1. *$\psi_{\mathbf{I},\mathbf{I},P_Z}$ is one-to-one;*

2. *$\psi_{\mathbf{I},\mathbf{I},P_Z}$ and its inverse are $C^\infty$;*

3. *$\forall y \in \mathbb{R}^n, \psi_{\mathbf{I},\mathbf{I},P_Z}(y)$ is the unique global minimum and unique stationary point of*

$$z \mapsto \frac{1}{2}\|y - \mathbf{I}z\|^2 + \phi(z), \text{ with:} \tag{4}$$

$$\phi(z) = \phi_{\mathbf{I},\mathbf{I},P_Z}(z) := \begin{cases} -\frac{1}{2}\|\psi_{\mathbf{I},\mathbf{I},P_Z}^{-1}(z) - z\|_2^2 - \log p_Y[\psi_{MMSE}^{-1}(z)]; & \text{for } z \in Im\psi_{\mathbf{I},\mathbf{I},P_Z}; \\ +\infty, & \text{for } x \notin Im\psi_{\mathbf{I},\mathbf{I},P_Z}; \end{cases}$$

4. *The penalty function $\phi_{\mathbf{I},\mathbf{I},P_Z}$ is $C^\infty$;*

5. *Any penalty function $\phi(z)$ such that $\psi_{\mathbf{I},\mathbf{I},P_Z}(y)$ is a stationary point of (4) satisfies $\phi(z) = \phi_{\mathbf{I},\mathbf{I},P_Z}(z) + C$ for some constant $C$ and all $z$.*

## 2.2 Non-white noise

Suppose now that $B \in \mathbb{R}^n$ is a centred non-degenerate normal Gaussian variable with a (positive definite) covariance matrix $\mathbf{\Sigma}$. Using a standard noise whitening technique, $\mathbf{\Sigma}^{-1/2} B \sim \mathcal{N}(0, \mathbf{I})$. This makes our denoising problem equivalent to $y_{\mathbf{\Sigma}} = z_{\mathbf{\Sigma}} + b_{\mathbf{\Sigma}}$, with $y_{\mathbf{\Sigma}} := \mathbf{\Sigma}^{-1/2} y$, $z_{\mathbf{\Sigma}} := \mathbf{\Sigma}^{-1/2} z$ and $b_{\mathbf{\Sigma}} := \mathbf{\Sigma}^{-1/2} b$, which is drawn from a Gaussian distribution with an identity covariance matrix. Finally, let $\|.\|_{\mathbf{\Sigma}}$ be the norm induced by the scalar product $\langle x, y \rangle_{\mathbf{\Sigma}} := \langle x, \mathbf{\Sigma}^{-1} y \rangle$.

**Corollary 1** (non-white Gaussian noise). *For any non-degenerate prior $P_Z$, any non-degenerate $\mathbf{\Sigma}$, $Y = Z + B$, we have:*

1. *$\psi_{\mathbf{I}, \mathbf{\Sigma}, P_Z}$ is one-to-one.*

2. *$\psi_{\mathbf{I}, \mathbf{\Sigma}, P_Z}$ and its inverse are $C^\infty$.*

3. *$\forall y \in \mathbb{R}^n, \psi_{\mathbf{I}, \mathbf{\Sigma}, P_Z}(y)$ is the unique global minimum and stationary point of*

$$z \mapsto \frac{1}{2} \|y - \mathbf{I}z\|_{\mathbf{\Sigma}}^2 + \phi_{\mathbf{I}, \mathbf{\Sigma}, P_Z}(z).$$

*with $\phi_{\mathbf{I}, \mathbf{\Sigma}, P_Z}(z) := \phi_{\mathbf{I}, \mathbf{I}, P_{\mathbf{\Sigma}^{-1/2} Z}}(\mathbf{\Sigma}^{-1/2} z)$*

4. *$\phi_{\mathbf{I}, \mathbf{\Sigma}, P_Z}$ is $C^\infty$.*

*As with white noise, up to an additive constant, $\phi_{\mathbf{I}, \mathbf{\Sigma}, P_Z}$ is the only penalty with these properties.*

*Proof.* First, we introduce a simple lemma that is pivotal throughout each step of this section.

**Lemma 1.** *For any function $f : \mathbb{R}^n \to \mathbb{R}$ and any $\mathbf{M} \in \mathbb{R}^{n \times p}$, we have:*

$$\mathbf{M} \operatorname*{argmin}_{v \in \mathbb{R}^p} f(\mathbf{M}v) = \operatorname*{argmin}_{u \in \mathrm{range}(\mathbf{M}) \subseteq \mathbb{R}^n} f(u).$$

Now, the linearity of the (conditional) expectation makes it possible to write

$$\mathbf{\Sigma}^{-1/2} \mathbb{E}(Z|Y = y) = \mathbb{E}(\mathbf{\Sigma}^{-1/2} Z | \mathbf{\Sigma}^{-1/2} Y = \mathbf{\Sigma}^{-1/2} y)$$
$$\Leftrightarrow \mathbf{\Sigma}^{-1/2} \psi_{\mathbf{I}, \mathbf{\Sigma}, P_Z}(y) = \psi_{\mathbf{I}, \mathbf{I}, P_{\mathbf{\Sigma}^{-1/2} Z}}(\mathbf{\Sigma}^{-1/2} y).$$

Using Theorem 1, it follows that:

$$\psi_{\mathbf{I}, \mathbf{\Sigma}, P_Z}(y) = \mathbf{\Sigma}^{1/2} \psi_{\mathbf{I}, \mathbf{I}, P_{\mathbf{\Sigma}^{-1/2} Z}}(\mathbf{\Sigma}^{-1/2} y)$$

From this property and Theorem 1, it is clear that $\psi_{\mathbf{I}, \mathbf{\Sigma}, P_Z}$ is one-to-one, $C^\infty$, as well as its inverse. Now, using Lemma 1 with $\mathbf{M} = \mathbf{\Sigma}^{1/2}$, we get:

$$\psi_{\mathbf{I}, \mathbf{\Sigma}, P_Z}(y) = \mathbf{\Sigma}^{1/2} \operatorname*{argmin}_{z' \in \mathbb{R}^n} \left\{ \frac{1}{2} \|\mathbf{\Sigma}^{-1/2} y - z'\|^2 + \phi_{\mathbf{I}, \mathbf{I}, P_{\mathbf{\Sigma}^{-1/2} Z}}(z') \right\}$$

$$= \operatorname*{argmin}_{z \in \mathbb{R}^n} \left\{ \frac{1}{2} \|\mathbf{\Sigma}^{-1/2} y - \mathbf{\Sigma}^{-1/2} z\|^2 + \phi_{\mathbf{I}, \mathbf{I}, P_{\mathbf{\Sigma}^{-1/2} Z}}(\mathbf{\Sigma}^{-1/2} z) \right\}$$

$$= \operatorname*{argmin}_{z \in \mathbb{R}^n} \left\{ \frac{1}{2} \|y - z\|_{\mathbf{\Sigma}}^2 + \phi_{\mathbf{I}, \mathbf{\Sigma}, P_Z}(z) \right\},$$

with $\phi_{\mathbf{I}, \mathbf{\Sigma}, P_Z}(z) := \phi_{\mathbf{I}, \mathbf{I}, P_{\mathbf{\Sigma}^{-1/2} Z}}(\mathbf{\Sigma}^{-1/2} z)$. This definition also makes it clear that $\phi_{\mathbf{I}, \mathbf{\Sigma}, P_Z}$ is $C^\infty$, and that this minimizer is unique (and is the only stationary point).

$\square$

## 2.3 A simple under-determined problem

As a step towards handling the more generic linear inverse problem $y = \boldsymbol{A}z + b$, we will investigate the particular case where $\boldsymbol{A} = [\mathbf{I} \ \ \mathbf{0}]$. For the sake of readability, for any two (column) vectors $u, v$, let us denote $[u; v]$ the concatenated (column) vector. First and foremost let us decompose the MMSE estimator into two parts, composed of the first $n$ and last $(D-n)$ components :

$$\psi_{[\mathbf{I} \ \ \mathbf{0}], \boldsymbol{\Sigma}, P_Z}(y) := [\psi_1(y); \psi_2(y)]$$

**Corollary 2** (simple under-determined problem). *For any non-degenerate prior $P_Z$, any non-degenerate $\boldsymbol{\Sigma}$, we have:*

1. $\psi_1(y) = \psi_{\mathbf{I}, \boldsymbol{\Sigma}, P_Z}(y)$ *is one-to-one and $C^\infty$. Its inverse and $\phi_{\mathbf{I}, \boldsymbol{\Sigma}, P_Z}$ are also $C^\infty$;*

2. $\psi_2(y) = (p_B \star g)(y)/(p_B \star P_Z)(y)$ *(with $g(z_1) := \mathbb{E}(Z_2|Z_1 = z_1)p(z_1)$) is $C^\infty$;*

3. $\psi_{[\mathbf{I} \ \ \mathbf{0}], \boldsymbol{\Sigma}, P_Z}$ *is injective.*

*Moreover, let $h : \mathbb{R}^{(D-n)} \times \mathbb{R}^{(D-n)} \mapsto \mathbb{R}^+$ be any function such that $h(x_1, x_2) = 0 \Rightarrow x_1 = x_2$,*

3. $\forall y \in \mathbb{R}^n, \psi_{[\mathbf{I} \ \ \mathbf{0}], \boldsymbol{\Sigma}, P_Z}(y)$ *is the unique global minimum and stationary point of*

$$z \mapsto \frac{1}{2}\|y - [\mathbf{I} \ \ \mathbf{0}]z\|_{\boldsymbol{\Sigma}}^2 + \phi^h_{[\mathbf{I} \ \ \mathbf{0}], \boldsymbol{\Sigma}, P_Z}(z)$$

*with $\phi^h_{[\mathbf{I} \ \ \mathbf{0}], \boldsymbol{\Sigma}, P_Z}(z) := \phi_{\mathbf{I}, \boldsymbol{\Sigma}, P_Z}(z_1) + h(z_2, \psi_2 \circ \psi_1^{-1}(z_1))$ and $z = [z_1; z_2]$.*

4. $\phi_{[\mathbf{I} \ \ \mathbf{0}], \boldsymbol{\Sigma}, P_Z}$ *is $C^\infty$ if and only if $h$ is.*

*Proof.* The expression of $\psi_2(y)$ is obtained by Bayes rule in the integral defining the conditional expectation. The smoothing effect of convolution with the Gaussian $p_B(b)$ implies the $C^\infty$ nature of $\psi_2$. Let $Z_1 = [\mathbf{I} \ \ \mathbf{0}]Z$. Using again the linearity of the expectation, we have:

$$[\mathbf{I} \ \ \mathbf{0}]\psi_{[\mathbf{I} \ \ \mathbf{0}], \boldsymbol{\Sigma}, P_Z}(y) = \mathbb{E}([\mathbf{I} \ \ \mathbf{0}]Z|Y = y) = \mathbb{E}(Z_1|Y = y) = \psi_{\mathbf{I}, \boldsymbol{\Sigma}, P_Z}(y).$$

Hence, $\psi_1(y) = \psi_{\mathbf{I}, \boldsymbol{\Sigma}, P_Z}(y)$. Given the properties of $h$, we have:

$$\psi_2(y) = \underset{z_2 \in \mathbb{R}^{(D-n)}}{\operatorname{argmin}} h\left(z_2, \psi_2 \circ \psi_1^{-1}(\psi_1(y))\right).$$

It follows that:

$$\psi_{[\mathbf{I} \ \ \mathbf{0}], \boldsymbol{\Sigma}, P_Z}(y) = \underset{z = [z_1; z_2] \in \mathbb{R}^D}{\operatorname{argmin}} \frac{1}{2}\|y - z_1\|_{\boldsymbol{\Sigma}}^2 + \phi_{\mathbf{I}, \boldsymbol{\Sigma}, P_Z}(z_1) + h(z_2, \psi_2 \circ \psi_1^{-1}(z_1)).$$

From the definitions of $\psi_{[\mathbf{I} \ \ \mathbf{0}], \boldsymbol{\Sigma}, P_Z}$ and $h$, it is clear, using Corollary 1 that $\psi_{[\mathbf{I} \ \ \mathbf{0}], \boldsymbol{\Sigma}, P_Z}$ is injective, is the unique minimizer and stationary point, and that $\phi_{[\mathbf{I} \ \ \mathbf{0}], \boldsymbol{\Sigma}, P_Z}$ is $C^\infty$ if and only if $h$ is. □

## 2.4 Inverse Problem

We are now equipped to generalize our result to an arbitrary full rank matrix $\boldsymbol{A}$. Using the *Singular Value Decomposition*, $\boldsymbol{A}$ can be factored as:

$$\boldsymbol{A} = \boldsymbol{U}[\boldsymbol{\Delta} \ \ \mathbf{0}]\boldsymbol{V}^\top = \tilde{\boldsymbol{U}}[\mathbf{I} \ \ \mathbf{0}]\boldsymbol{V}^\top, \text{ with } \tilde{\boldsymbol{U}} = \boldsymbol{U}\boldsymbol{\Delta}.$$

Our problem is now equivalent to $y' := \tilde{\boldsymbol{U}}^{-1}y = [\mathbf{I} \ \ \mathbf{0}]\boldsymbol{V}^\top z + \tilde{\boldsymbol{U}}^{-1}b =: z' + b'$.

Let $\tilde{\boldsymbol{\Sigma}} = \tilde{\boldsymbol{U}}^{-1}\boldsymbol{\Sigma}\tilde{\boldsymbol{U}}^{-\top}$. Note that $B' \sim \mathcal{N}(0, \tilde{\boldsymbol{\Sigma}})$.

**Theorem 2** (Main result). *Let $h : \mathbb{R}^{(D-n)} \times \mathbb{R}^{(D-n)} \mapsto \mathbb{R}^+$ be any function such that $h(x_1, x_2) = 0 \Rightarrow x_1 = x_2$. For any non-degenerate prior $P_Z$, any non-degenerate $\boldsymbol{\Sigma}$ and $\boldsymbol{A}$, we have:*

1. $\psi_{\boldsymbol{A}, \boldsymbol{\Sigma}, P_Z}$ *is injective.*

2. $\forall y \in \mathbb{R}^n, \psi_{[\mathbf{I}\ \mathbf{0}],\boldsymbol{\Sigma},P_Z}(y)$ *is the unique global minimum and stationary point of*
   $z \mapsto \frac{1}{2}\|y - \boldsymbol{A}z\|_{\boldsymbol{\Sigma}}^2 + \phi_{\boldsymbol{A},\boldsymbol{\Sigma},P_Z}^h(z)$, *with* $\phi_{\boldsymbol{A},\boldsymbol{\Sigma},P_Z}^h(z) := \phi_{[\mathbf{I}\ \mathbf{0}],\tilde{\boldsymbol{\Sigma}},P_{\boldsymbol{V}^\top Z}}^h(\boldsymbol{V}^\top z)$.

3. $\phi_{\boldsymbol{A},\boldsymbol{\Sigma},P_Z}$ *is* $C^\infty$ *if and only if* $h$ *is.*

*Proof.* First note that:

$$\boldsymbol{V}^\top \psi_{\boldsymbol{A},\boldsymbol{\Sigma},P_Z}(y) = \boldsymbol{V}^\top \mathbb{E}(Z|Y=y) = \mathbb{E}(Z'|Y'=y') = \psi_{[\mathbf{I}\ \mathbf{0}],\tilde{\boldsymbol{\Sigma}},P_{Z'}}$$

$$= \underset{z'}{\arg\min} \frac{1}{2}\|\boldsymbol{U}^\top y - [\mathbf{I}\ \mathbf{0}]z'\|_{\tilde{\boldsymbol{\Sigma}}}^2 + \phi_{[\mathbf{I}\ \mathbf{0}],\tilde{\boldsymbol{\Sigma}},P_{\boldsymbol{V}^\top Z}}^h(z'),$$

using Corollary 2. Now, using Lemma 1, we have:

$$\psi_{\boldsymbol{A},\boldsymbol{\Sigma},P_Z}(y) = \underset{z}{\arg\min} \frac{1}{2}\|\boldsymbol{U}^\top\left(y - \boldsymbol{U}[\mathbf{I}\ \mathbf{0}]\boldsymbol{V}^\top\right)\|_{\tilde{\boldsymbol{\Sigma}}}^2 + \phi_{[\mathbf{I}\ \mathbf{0}],\tilde{\boldsymbol{\Sigma}},P_{\boldsymbol{V}^\top Z}}^h(\boldsymbol{V}^\top z)$$

$$= \underset{z}{\arg\min} \frac{1}{2}\|y - \boldsymbol{A}z\|_{\boldsymbol{\Sigma}}^2 + \phi_{[\mathbf{I}\ \mathbf{0}],\tilde{\boldsymbol{\Sigma}},P_{\boldsymbol{V}^\top Z}}^h(\boldsymbol{V}^\top z)$$

The other properties come naturally from those of Corollary 2. $\qquad\square$

*Remark* 1. If $\boldsymbol{A}$ is invertible (hence square), $\psi_{\boldsymbol{A},\boldsymbol{\Sigma},P_Z}$ is one-to-one. It is also $C^\infty$, as well as its inverse and $\phi_{\boldsymbol{A},\boldsymbol{\Sigma},P_Z}$.

# 3 Connections between the MMSE and regularization-based estimators

Equipped with the results from the previous sections, we can now have a clearer idea about how MMSE estimators, and those produced by a regularization-based approach relate to each other. This is the object of the present section.

## 3.1 Obvious connections

Some simple observations of the main theorem can already shed some light on those connections. First, for any prior, and as long as $\boldsymbol{A}$ is invertible, we have shown that there exists a corresponding regularizing term (which is unique up to an additive constant). This simply means that the set of MMSE estimators in linear inverse problems with Gaussian noise is a subset of the set of estimators that can be produced by a regularization approach with a quadratic data-fitting term.

Second, since the corresponding penalty is necessarily smooth, it is in fact only a *strict* subset of such regularization estimators. In other words, for some regularizers, there cannot be any interpretation in terms of an MMSE estimator. For instance, as pinpointed by [8], all the non-$C^\infty$ regularizers belong to that category. Among them, all the sparsity-inducing regularizers (the $\ell^1$ norm, among others) fall into this scope. This means that when it comes to solving a linear inverse problem (with an invertible $\boldsymbol{A}$) under Gaussian noise, sparsity inducing penalties are necessarily *suboptimal* (in a mean squared error sense).

## 3.2 Relating desired computational properties to the evidence

Let us now focus on the MMSE estimators (which also can be written as regularization-based estimators). As reported in the introduction, one of the reasons explaining the success of the optimization-based approaches is that one can have a better control on the computational efficiency on the algorithms via some appealing properties of the functional to minimize. An interesting question then is: can we relate these properties of the regularizer to the Bayesian priors, when interpreting the solution as an MMSE estimate?

For instance, when the regularizer is separable, one may easily rely on coordinate descent algorithms [9]. Here is a more formal definition:

**Definition 1** (Separability). *A vector-valued function* $f : \mathbb{R}^n \to \mathbb{R}^n$ *is* separable *if there exists a set of functions* $f_1, \ldots, f_n : \mathbb{R} \to \mathbb{R}$ *such that* $\forall x \in \mathbb{R}^n, f(x) = (f_i(x_i))_{i=1}^n$.

A scalar-valued function $g : \mathbb{R}^n \to \mathbb{R}$ *is* additively separable *(resp.* multiplicatively separable*) if there exists a set of functions* $g_1, \ldots, g_n : \mathbb{R} \to \mathbb{R}$ *such that* $\forall x \in \mathbb{R}^n, g(x) = \sum_{i=1}^n g_i(x_i)$ *(resp.* $g(x) = \prod_{i=1}^n g_i(x_i)$*).*

Especially when working with high-dimensional data, coordinate descent algorithms have proven to be very efficient and have been extensively used for machine learning [10, 11].

Even more evidently, when solving optimization problems, dealing with convex functions ensures that many algorithms will provably converge to the global minimizer [6]. As a consequence, it would be interesting to be able to characterize the set of priors for which the MMSE estimate can be expressed as a minimizer of a convex function.

The following lemma precisely addresses these issues. For the sake of simplicity and readability, we focus on the specific case where $A = I$ and $\Sigma = I$.

**Lemma 2** (Convexity and Separability). *For any non-degenerate prior* $P_Z$*, Theorem 1 says that* $\forall y \in \mathbb{R}^n, \psi_{\mathbf{I},\mathbf{I},P_Z}(y)$ *is the unique global minimum and stationary point of* $z \mapsto \frac{1}{2}\|y - \mathbf{I}z\|^2 + \phi_{\mathbf{I},\mathbf{I},P_Z}(z)$*. Moreover, the following results hold:*

1. $\phi_{\mathbf{I},\mathbf{I},P_Z}$ *is convex if and only if* $p_Y(y) := p_B \star P_Z(y)$ *is log-concave,*

2. $\phi_{\mathbf{I},\mathbf{I},P_Z}$ *is additively separable if and only if* $p_Y(y)$ *is multiplicatively separable.*

*Proof of Lemma 2.* From Lemma II.1 in [8], the Jacobian matrix $J[\psi_{\mathbf{I},\mathbf{I},P_Z}](y)$ is positive definite hence invertible. Derivating $\phi_{\mathbf{I},\mathbf{I},P_Z}[\psi_{\mathbf{I},\mathbf{I},P_Z}(y)]$ from its definition in Theorem 1, we get:

$$J[\psi_{\mathbf{I},\mathbf{I},P_Z}](y)\nabla\phi_{\mathbf{I},\mathbf{I},P_Z}[\psi_{\mathbf{I},\mathbf{I},P_Z}(y)]$$

$$= \nabla\left[-\frac{1}{2}\|y - \psi_{\mathbf{I},\mathbf{I},P_Z}(y)\|_2^2 - \log p_Y(y)\right]$$

$$= -(\mathbf{I}_n - J[\psi_{\mathbf{I},\mathbf{I},P_Z}](y))(y - \psi_{\mathbf{I},\mathbf{I},P_Z}(y)) - \nabla\log p_Y(y)$$

$$= (\mathbf{I}_n - J[\psi_{\mathbf{I},\mathbf{I},P_Z}](y))\nabla\log p_Y(y) - \nabla\log p_Y(y)$$

$$= -J[\psi_{\mathbf{I},\mathbf{I},P_Z}](y)\nabla\log p_Y(y)$$

Then:

$$\nabla\phi_{\mathbf{I},\mathbf{I},P_Z}[\psi_{\mathbf{I},\mathbf{I},P_Z}(y)] = -\nabla\log p_Y(y).$$

Derivating this expression once more, we get:

$$J[\psi_{\mathbf{I},\mathbf{I},P_Z}](y)\nabla^2\phi_{\mathbf{I},\mathbf{I},P_Z}[\psi_{\mathbf{I},\mathbf{I},P_Z}(y)] = -\nabla^2\log p_Y(y).$$

Hence:

$$\nabla^2\phi_{\mathbf{I},\mathbf{I},P_Z}[\psi_{\mathbf{I},\mathbf{I},P_Z}(y)] = -J^{-1}[\psi_{\mathbf{I},\mathbf{I},P_Z}](y)\nabla^2\log p_Y(y).$$

As $\psi_{\mathbf{I},\mathbf{I},P_Z}$ is one-to-one, $\phi_{\mathbf{I},\mathbf{I},P_Z}$ is convex if and only if $\phi_{\mathbf{I},\mathbf{I},P_Z}[\psi_{\mathbf{I},\mathbf{I},P_Z}]$ is. It also follows that:

$$\phi_{\mathbf{I},\mathbf{I},P_Z} \text{ convex} \Leftrightarrow \nabla^2\phi_{\mathbf{I},\mathbf{I},P_Z}[\psi_{\mathbf{I},\mathbf{I},P_Z}(y)] \succcurlyeq 0$$

$$\Leftrightarrow -J^{-1}[\psi_{\mathbf{I},\mathbf{I},P_Z}](y)\nabla^2\log p_Y(y) \succcurlyeq 0$$

As $J[\psi_{\mathbf{I},\mathbf{I},P_Z}](y) = \mathbf{I}_n + \nabla^2\log p_Y(y)$, the matrices $\nabla^2\log p_Y(y)$, $J[\psi_{\mathbf{I},\mathbf{I},P_Z}](y)$, and $J^{-1}[\psi_{\mathbf{I},\mathbf{I},P_Z}](y)$ are simultaneously diagonalisable. It follows that the matrices $J^{-1}[\psi_{\mathbf{I},\mathbf{I},P_Z}](y)$ and $\nabla^2\log p_Y(y)$ commute. Now, as $J[\psi_{\mathbf{I},\mathbf{I},P_Z}](y)$ is positive definite, we have:

$$-J^{-1}[\psi_{\mathbf{I},\mathbf{I},P_Z}](y)\nabla^2\log p_Y(y) \succcurlyeq 0 \Leftrightarrow \nabla^2\log p_Y(y) \preccurlyeq 0.$$

It follows that $\phi_{\mathbf{I},\mathbf{I},P_Z}$ is convex if and only if $p_Y(y) := p_B \star P_X(y)$ is log-concave.

By its definition (II.3) in [8], it is clear that:

$$\phi_{\mathbf{I},\mathbf{I},P_Z} \text{ is additively separable} \Leftrightarrow \psi_{\mathbf{I},\mathbf{I},P_Z} \text{ is separable.}$$

Using now equation (II.2) in [8], we have:

$$\psi_{\mathbf{I},\mathbf{I},P_Z} \text{ is separable} \Leftrightarrow \nabla\log p_Y \text{ is separable}$$

$$\Leftrightarrow \log p_Y \text{ is additively separable}$$

$$\Leftrightarrow p_Y \text{ is multiplicatively separable.}$$

$\square$

*Remark* 2. This lemma focuses on the specific case where $\boldsymbol{A} = \mathbf{I}$ and a white Gaussian noise. However, considering the transformations induced by a shift to an arbitrary invertible matrix $\boldsymbol{A}$ and to an arbitrary non-degenerate covariance matrix $\boldsymbol{\Sigma}$, which are depicted throughout Section 2, it is easy to see that the result on convexity carries over. An analogous (but more complicated) result could be also derived regarding separability. We leave that part to the interested reader.

These results provide a precise characterization of conditions on the Bayesian priors so that the MMSE estimator can be expressed as minimizer of a convex or separable function. Interestingly, those conditions are expressed in terms of the probability distribution function (*pdf* in short) of the observations $p_Y$, which is sometimes referred to as the *evidence*. The fact that the evidence plays a key role in Bayesian estimation has been observed in many contexts, see for example [12]. Given that we assume that the noise is Gaussian, its pdf $p_B$ always is log-concave. Thanks to a simple property of the convolution of log-concave functions, it is sufficient that the prior on the signal $p_Z$ is log-concave to ensure that $p_Y$ also is. However, it is *not* a necessary condition. This means that there are some priors $p_X$ that are *not* log-concave such that the associated MMSE estimator can still be expressed as the minimizer of a functional with a convex regularizer. For a more detailed analysis of this last point, in the specific context of Bernoulli-Gaussian priors (which are not log-concave), please refer to the technical report [13].

From this result, one may also draw an interesting negative result. If the distribution of the observation $y$ is *not* log-concave, then, the MMSE estimate *cannot* be expressed as the solution of a convex regularization-oriented formulation. This means that, with a quadratic data-fitting term, a convex approach to signal estimation *cannot* be optimal (in a mean squared error sense).

## 4   Prospects

In this paper we have extended a result, stating that in the context of linear inverse problems with Gaussian noise, for any Bayesian prior, there exists a regularizer $\phi$ such that the MMSE estimator can be expressed as the solution of regularized regression problem (2). This result is a generalization of a result in [8]. However, we think it could be extended with regards to many aspects. For instance, our proof of the result naturally builds on elementary bricks that combine in a way that is imposed by the definition of the linear inverse problem. However, by developing more bricks and combining them in different ways, it may be possible to derive analogous results for a variety of other problems.

Moreover, in the situation where $\boldsymbol{A}$ is not invertible (i.e. the problem is under-determined), there is a large set of admissible regularizers (i.e. up to the choice of a function $h$ in Corollary 2). This additional degree of freedom might be leveraged in order to provide some additional desirable properties, from an optimization perspective, for instance.

Also, our result relies heavily on the choice of a quadratic loss for the data-fitting term and on a Gaussian model for the noise. Naturally, investigating other choices (e.g. logistic or hinge loss, Poisson noise, to name a few) is a question of interest. But a careful study of the proofs in [8] suggests that there is a peculiar connection between the Gaussian noise model on the one hand and the quadratic loss on the other hand. However, further investigations should be conducted to get a deeper understanding on how these really interplay on a higher level.

Finally, we have stated a number of negative results regarding the non-optimality of sparse decoders or of convex formulations for handling observations drawn from a distribution that is not log-concave. It would be interesting to develop a metric in the estimators space in order to quantify, for instance, how "far" one arbitrary estimator is from an optimal one, or, in other words, what is the intrinsic cost of convex relaxations.

## Acknowledgements

This work was supported in part by the European Research Council, PLEASE project (ERC-StG-2011-277906).

## Footnotes

[1]which is the Bayesian optimal estimator in a 0/1 loss sense, for discrete signals.

[2]We only need to assume that $Z$ does not intrinsically live almost surely in a lower dimensional hyperplane. The results easily generalize to this degenerate situation by considering appropriate projections of $y$ and $z$. Similar remarks are in order for the non-degeneracy assumptions on $\boldsymbol{\Sigma}$ and $\boldsymbol{A}$.

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
