[Reviews · NeurIPS 2013]

Submitted by Assigned_Reviewer_4

The paper presents some connections and highlights fundamentals differences between the "Bayesian approach" and the "penalized regression" approach. If the two approaches are heavily used to "solve" inverse problems, they corresponds to two different "philosophy", and can leads to long and fruitless debates with poor arguments as : "my method is the best". For example, the number of results given by google with "Bayesian church" as keywords is in fact hilarious !

Here, the authors try to understand what is "behind" the two approaches. In particular, the paragraph on the "MAP" (miss)interpretation of regularization approach is very instructive. However, beyond the bird's eye view on the two approaches, there is very important theoretical results stating great connections, but also some incompatibilities.

My only "criticism" is that one wants more : are we able to find some concrete examples of functional whose minimizer corresponds to a MMSE estimates ?
Even if the convert is not true, are we able to estimate the distance between a MMSE estimates and the minimizer of a functional ? I am particularly thinking of how far are we from the MMSE estimates of a Bernoulli-Gaussian prior with the L1 solution for sparse estimation.
Summary: This one of the most pleasant I have had to review since a long time. The stated results are very important for people who study inverse problems, and open new research directions.

Submitted by Assigned_Reviewer_5

This paper establishes that all Bayes estimates of z under squared error loss (i.e., posterior mean) in the model

y ~ N(Az, \Sigma)
z ~ P

correspond to MAP estimates (i.e., penalized regression) which optimize

||y - Az||_2^2 + \phi(z)

for some \phi depending on A, P, and \Sigma. This is an extension of Gribonval (2012), which established this result in the case of A = \Sigma = Id. The author also characterizes the Bayesian problems for which \phi ends up being a convex regularizer.

Although the proofs are entirely straightforward given Gribonval (2012), the result is interesting, and the extension to A non-orthogonal is significant (the extension to correlated errors is less interesting). Furthermore, the 2012 paper was published in the IEEE Trans. on Signal Processing, and I think that this result would be of interest to the machine learning community, a large segment of which does not follow the signal processing literature.

However, I would note a few things: the fact that the prior \phi depends on A is somewhat unsatisfactory, as it suggests the prior somehow depends on the likelihood. Also, there seem to be many typos, and I hope the author will carefully proofread the manuscript before the final submission. I also think that the author should be careful in saying that "sparsity inducing penalties are necessarily suboptimal" in Section 3.1. They are only suboptimal if you assume the truth has to be generated from some prior! Given the wealth of literature around the optimality of L1 regularization over linear estimates (which do correspond to an MMSE estimator) over certain function classes, I think this is a dangerous statement to make, even if it is qualified by "in a mean squared error sense".

Editorial suggestions:
- The term is "quid pro quo," not "qui pro quo."
- Line 76: "A qui pro quo been these..."?
- Line 88: "ping" should be "pinv"
- Line 303-4: should be "in terms of"
- Line 312: should be "one of the reasons"
- Line 414: "investigating other choices"
Summary: This is a solid paper that addresses an interesting topic in machine learning: the connection between Bayesian and MAP/frequentist approaches. Although the theoretical contribution is not the most technical or original, especially given Gribonval (2012), I think that the general interest of this topic to the community justifies acceptance of this paper.

Submitted by Assigned_Reviewer_6

This paper extends previous work on the relationship between MAP and MMSE estimators in the context of linear inverse problems under Gaussian noise. After pointing out what the authors claim to be common misunderstandings (more about this later), the core of the paper is about establishing the following: if p(z) is a prior (under some conditions) for which the MMSE estimate exists, then there is another prior such that the MAP estimate coincides with the MMSE estimate under the first prior. The paper is quite well written and clear (with the exception noted below) and the technical part is quite interesting and solid; it is relevant work, although it is essentially an extension of previous work.
Summary: My main concern about this paper does not relate to its technical contents, but to its "semantics", that is, how the author interpret their results, in the light of Bayesian point estimation theory. In fact, I think that Section 1.2 is misleading enough to prevent me from recommending acceptance of this paper in its current form.

Concerning the technical body of the paper, I believe the results are technically solid and relevant, and deserve eventual publication; however, the paper is somewhat spoiled by a misleading motivation and interpretation of the results.

First of all, the authors seem to imply that the MMSE criterion (1) is Bayesian, while criterion (2) is not. Of course, MAP/regularization is as Bayesian as MMSE; they simply correspond do different losses. While the conditional mean (MMSE) minimizes the posterior expectation of the squared error loss, the MAP estimate minimizes the posterior expectation of the 0/1 loss. In fact, equation (2) is also a Bayesian point estimator, assuming a prior of the form p(x) = K exp(-phi(z), where K is a normalization constant independent of z, thus irrelevant in (2).

Section 1.2 is full of misunderstandings or, at least, several misleading statements. First, it is absolutely true that the optimization problem (2) with regularizer phi(z) is indeed the MAP point estimate with prior exp(-phi(z)); this is simply an equality, which has nothing to do with interpretations. It is obvious that z_MAP = argmax_z ( exp(-|| y - A z ||_2^2) exp(- phi(z)) ) = argmin_z ( || y - A z ||_2^2 + phi(z)), which is exactly (2). Whether this is a good estimate or not, that is another issue, and it essentially boils down to deciding if the 0/1 loss is the one that makes sense for a given application/scenario. Of course, if one chooses the 0/1 loss, that is, the MAP estimate (given by (2)), then it only makes sense to measure its quality in terms of the expected value of the 0/1 loss. Consequently, the example given of the Laplacian distributed z is misleading, because the performance is being measured using the MSE (that is, the squared error loss), while the MAP is optimal for the 0/1 loss, so there is an obvious mismatch between the loss for which the estimator is optimized and the one used to measure its performance. Of course, the MMSE "decoder" under the Laplacian prior would be the Bayes-optimal thing to do, and it does not yield sparse estimates.

In the second paragraph of this section, the authors insist on their erroneous statement: "It is a common mistake, for instance, to consider that the Lasso is a method that computes a MAP estimate with a Laplacian prior." It is not a mistake; as shown above, it is a trivial equality.

Minor issues: In the sentence "A qui pro quo been these distinct notions
of priors has crystallized...", the word "been" seems to be wrong; should it be "between"? What is ping(A)? Is it MATLAB's pinv(A)?

Author Feedback

Author rebuttal: First, we would like to thank the reviewers for their fruitful remarks and their careful reading of our paper.
As a general remark, we also want to thank the reviewers for pointing out some of the numerous typos that were left in our manuscript.
The more specific answers we provide may be summarized as follows.
- To reviewer 1:
It is still unclear to us how to provide a practical example of a functional, whose minimizer corresponds to a MSSE estimate, beyond the expressions given in the main theorem and corollaries.
However, as for the issue raised about the distance between a MSSE estimate and a minimizer, and as stated in our conclusion, these are among the main issues we would like to investigate further.
Regarding the special case of a Bernoulli-Gaussian prior, we invite you to check out the technical report (http://hal.inria.fr/hal-00751996).
- To reviewer 2:
The dependence on A in the regularizing term, if unsatisfactory, directly comes from the dependency, in the whole analysis (as pointed out by [Raphan & Simoncelli 07]), on the evidence of the data.
On a side note, this dependency is reminiscent of [Wipf & Rao 11], where the authors suggest the use of non-separable penalties (depending on A) for sparse estimation.
Regarding the non-optimality of sparse estimators, we are not sure to understand what the reviewer is exactly referring to but certainly would like some pointers.
However, we do agree that this non-optimality is only to be understood in a mean squared error sense (but for any prior on the data) as sparse estimators can be proven optimal under different criteria.
- To reviewer 3:
A longer discussion regarding the nature of Bayesian estimators, on which our paper relies, was provided by [Gribonval 12].
In order to focus on the novelty of the present paper, we had decided to skip most of it in our work.
Of course the MAP estimator (corresponding to the 0/1 loss) is "as Bayesian" as the MMSE (with the quadratic loss), as pointed out by reviewer 3.
This being said, the 0/1 criterion can seem unnatural for many tasks and applications, which may explain why the mean squared error (hence the MMSE) can be often favored.
Regarding the interpretation, our main point was to state that penalized least squares *need not be* considered as a MAP estimation and can be also seen (as our paper shows) as yielding MMSE estimates (for certain C^\infty penalties) with some prior, which is different from the "MAP prior".
In particular, we have had many occasions to hear colleagues, commenting on the use of an l1 penalty, argue that "since this is the MAP with a Laplacian distribution", "it should be tested on Laplacian distributed data" or "it is ideally tailored for Laplacian data".
We wanted to clearly state that this point, for instance, is erroneous as the multiple "Bayesian interpretations" of penalized least squares precisely show that the choice of a penalty does *NOT* uniquely correspond to the choice of a specific prior on the data at hand.
In other words, one implicitly makes an assumption on the distribution of the data only when one jointly chooses a penalty term *AND* a relevant loss function for the task at hand (which often is non-trivial).
We plan to carefully update our wording in the final version to remove possible misunderstandings.